

# Elemental composition, iron mineralogy and solubility of anthropogenic and natural mineral dust aerosols in Namibia: a case study analysis from the AEROCLO-sA campaign

Paola Formenti[1*], Chiara Giorio[2,3], Karine Desboeufs[1], Alexander Zherebker[2], Marco Gaetani[4], Clarissa Baldo[5], Gautier Landrot[6], Simona Montebello[2,7], Servanne Chevaillier[1], Sylvain Triquet[1], Guillaume Siour[5], Claudia Di Biagio[1], Francesco Battaglia[1], Jean-François Doussin[5], Anais Feron[1,$], Andreas Namwoonde[8], and Stuart Piketh[9]

[1] Université Paris Cité and Univ. Paris Est Creteil, CNRS, LISA, F−75013 Paris, France
[2] Yusuf Hamied Department of Chemistry, University of Cambridge, Lensfield Road, Cambridge, CB2 1EW, United Kingdom
[3] Dipartimento di Scienze Chimiche, Università degli Studi di Padova, 35131 Padova, Italy
[4] Classe di Scienze Tecnologie e Società, Scuola Universitaria Superiore IUSS, 27100, Pavia, Italia
[5] Univ Paris Est Creteil and Université Paris Cité, CNRS, LISA, F−94010 Créteil, France
[6] Synchrotron SOLEIL, L'Orme des Merisiers, Saint-Aubin, France
[7] Department of Engineering 'Enzo Ferrari', University of Modena and Reggio Emilia, 41125 Modena, Italy
[8] Sam Nujoma Marine and Coastal Resources Research Centre, University of Namibia, Henties Bay, Namibia
[9] NorthWest University, Potchefstroom, South Africa
[$] now at INRAE

*Correspondence to: paola.formenti@lisa.ipsl.fr

## Abstract

This paper presents the results of three weeks of aerosol sampling at the Henties Bay coastal site in Namibia during the Aerosols, Radiation and Clouds in southern Africa (AEROCLO-sA) field campaign in August-September 2017. The campaign coincided with a transition period between two synoptic regimes and corresponded to a significant change in the aerosol composition measured at the site and in particular of that of mineral dust. During August, the dust was natural windblown from the southerly gravel plains with a composition consistent with that previously observed in Namibia. In September, the dust was fugitive from anthropogenic mining and possibly minor contribution of smelting emissions in northern Namibia or as far as the Copper Belt in Zambia, one of the regional hotspot of pollution.





Chemical analysis of filter samples highlights the difference in elemental composition, in
particular heavy metals, such as As, Cu, Cd, Pb, and Zn, but also silicon, in the anthropogenic
dust. The metal solubility of the natural dust was higher, including that of iron. In addition to
the higher content of iron oxides and the larger size of particles in the anthropogenic dust, we
found that the iron solubility, and, more in general, the metals' solubility, correlated to the high
concentrations of fluoride ion which are attributed to marine emissions from the Namibian shelf.
These results highlight in a renewed manner the importance of ocean-atmosphere exchanges
affecting both the atmospheric composition and the marine biogeochemistry in the Benguela
region.

## 1    Introduction

Mineral dust is an abundant component of the global atmosphere (Kok et al., 2023). Dust
particles in the atmosphere are released by the natural wind erosion of natural arid and semi-
arid areas of the globe. However, global dust emissions are also contributed by anthropogenic
activities such as labouring of bare soils for agriculture, pasture or construction, but also
fugitive dust from mining and road traffic activities (Knippertz and Stuut, 2014). Mineral dust is
a strong regulator of the Earth's climate and environment (Kok et al., 2023). In the atmosphere,
it contributes to both the direct and indirect radiative effects on climate by scattering and
absorbing solar and terrestrial radiation and forming cloud droplets in the liquid and ice phases
(Kok et al., 2023). It also affects the atmospheric composition and oxidative capacity by acting
as a source or a reactive sink of species from the gas phase (Usher et al., 2003). It also acts
as an irritating agent for the upper respiratory system and a vector of bacteria and infections
(Adebiyi et al., 2023). By deposition, mineral dust can provide nutrients and pollutants to the
sea water, changing the ocean's primary production (Knippertz and Stuut, 2014).
These considerations apply to the west coast of southern Africa, and Namibia in particular, a
hyper-arid climate where many dust sources co-exist (Vickery et al., 2013). Natural mineral
dust is emitted from coastal riverine sources, salty pans such as the Etosha and large gravel
plains ubiquitous around the country (Vickery et al., 2013; Dansie et al., 2017; Von Holdt et al.,
2018; Klopper et al., 2020; Shikwambana and Kganyago, 2022; Desboeufs et al., 2024). These
sources are active throughout the year as emissions occur under various wind regimes (Von
Holdt et al., 2018). Natural mineral dust from Namibia is transported within the shallow
boundary layer but being able to reach as far as Eastern Antarctica through long-range
transport (Gili et al., 2022). Previous research in Namibia has shown that natural mineral dust
from the coastal riverbeds and the gravel plains might contribute to oceanic productivity,
particularly along the coast (Dansie et al., 2022). This research also pointed to iron as a highly
soluble element in both the soil and windblown aerosol fraction and the control for the impact
of dust on oceanic productivity (Dansie et al., 2017a; 2017b; 2018; Desboeufs et al., 2024).



Furthermore, coastal pollution is an emerging issue of the present-day world (Strain et al.,
2022). Increased human activities and coastal developments are quickly affecting the air
quality but also the aquatic environment and biodiversity (Micella et al., 2024). Indeed, and
despite its low population, Namibia also has intense and emerging economic activities such as
mining (various heavy elements, including uranium; Mileusnić et al., 2014; Sracek, 2015;
Liebenberg-Enslin et al., 2020) and marine traffic transporting merchandise along the coast of
Africa and towards South America (Tournadre, 2014; Klopper et al., 2020). These activities
release fugitive dust from the mine locations as well as from the numerous road constructions
from and to the major national harbour, Walvis Bay (https://mwt.gov.na/projects; last accessed
26/11/2024). In Namibia, the accumulation of heavy metals in the shore and coastal waters
due to coastal mining (Onjefu et al., 2020) has been previously documented (Sylvanus et al.,
2016; Omoregie et al., 2019; Nekhoroshkov et al., 2021). Furthermore, in the austral
wintertime, Namibia is affected by anti-cyclonic circulation, resulting in the transport of light-
absorbing particles, likely from forest fires and mining areas such as the Zambian Copper Belt
(Formenti et al., 2018; Aurélien et al., 2022; Martinez-Alonso et al., 2023; Kříbek et al., 2023).
The composition of these emissions is little characterised to date while having the potential for
alter the oceanic productivity and microbial biogeochemistry (Adriano, 2001; Jordi et al., 2012;
Mahowald et al., 2018; Yang et al. 2019).
In this paper, we present a case study analysis of the differences and similarities of the
composition of natural and anthropogenic mineral dust sampled during the ground-based field
campaign of the Aerosols, Radiation and Clouds in southern Africa (AEROCLO-sA) project
(Formenti et al., 2019). The campaign was conducted in August-September 2017 in Henties
Bay (22°6′S, 14°30′E; 20 m above mean sea level) along the Namibian coast.
Based on analysis of the chemical composition and meteorological fields, we demonstrate the
origin of the anthropogenic dust and contrast its elemental composition, iron mineralogy and
solubility, and the type of organic matter with respect to that of natural dust measured at the
beginning of the campaign. Our analysis focusses on the iron mineralogy and solubility but
includes, for the first time the evaluation of the solubility of heavy metals transported with these
emissions.

## 2   Experimental

The AEROCLO-sA field campaign took place from 21 August to 13 September 2017 at the
Sam Nujoma Marine and Coastal Resources Research Centre (SANUMARC) of the University
of Namibia at Henties Bay (Formenti et al., 2019). This sampling site, operated on the long-
term as described by Formenti et al. (2018) and Klopper et al. (2020), was augmented with the



PortablE Gas and Aerosol Sampling UnitS (PEGASUS; https://pegasus.aeris-data.fr/; last
accessed 24/01/2025) mobile facility for the time of the campaign.
The PEGASUS facility consists of two marine containers (20-feet long) customized and
equipped for atmospheric research (Formenti et al., 2019). Air sampling is performed with two
high-volume aerosol inlets delivering approximately 450 L min$^{-1}$ each. At wind speeds between
5 to 10 m s$^{-1}$, typical for coastal Namibia (see Figure S1), sampling is almost isokinetic for
particles up to 40 µm in aerodynamic diameter (Rajot et al., 2008), which hereafter we named
total suspended particulate (TSP). The total sampled flow rate is distributed to online analysers
and to multiple- and single-stack sample collection units for off-line analysis of the bulk and
size-resolved chemical and mineralogical composition, soluble fraction and mixing state. The
details of the online instrumentation relevant to this publication are listed in Table S1 in the
supplementary material.

### 2.1   Sample collection

During the campaign, aerosol samples were collected both during day (approximately 07:00-
17:00 UTC) and night time (approximately 17:30-06:30 UTC). The sampling duration was
marginally adapted in real-time to the nature and the aerosol load of air masses using the
readings of the local wind speed and direction and of the aerosol mass concentration, also
measured online.
Four custom-made filter holders were used in parallel for collecting aerosols in the TSP
fraction. These were loaded with (i) one Teflon filter (Zefluor®, 2-µm pore size diameter, 47-
mm filter diameter); (ii) two polycarbonate membranes (Nuclepore®, 0.4-µm pore size
diameter, 37-mm and 47-mm filter diameter, respectively), and (iii) a quartz filter (Pall,
2500QAT-UP Tissuquartz, 47-mm filter diameter). The average sampling flow rate varied
between 20 and 30 L min$^{-1}$.
Two samples of the composition of particles smaller than 1 µm in diameter (hereafter named
PM$_1$ size fraction) were collected in parallel using two 4-stage Dekati® PM$_{10}$ Impactors, both
operated at 10 L min$^{-1}$. For these two samplers we used 25-mm polycarbonate membranes on
three impactor stages (> 10 µm, 10-2.5 µm and 2.5-1 µm) while the final filter stage, where the
PM$_1$ fraction is collected, was a polycarbonate membrane (Nuclepore®, 0.4-µm pore size, 47-
mm filter diameter) and a quartz filter (Pall, 2500QAT-UP Tissuquartz, 47-mm filter diameter),
respectively.
Before the campaign, Teflon and quartz membranes were cleaned for sampling organic
aerosols. Teflon membranes were rinsed with dichloromethane and baked at 100°C for 10
minutes. Quartz membranes were baked at 550°C for 12 hours. Both were conditioned in pre-
baked aluminium foils. The polycarbonate membranes were used for measuring the inorganic



and water-soluble fraction composition. The 37-mm and 25-mm membranes were used as
purchased, while the 47-mm ones were acid-washed according to the protocol described in
Desboeufs et al. (2024). All material was sealed and opened only before collection.
Immediately after exposure, all samples were sealed and stored at -18°C in the deep-freezer
available in the PEGASUS facility, from which they were transported back to the laboratory.
TSP filter samples were collected between 21 August and 12 September 2017, while PM$_1$
samples were collected from 26 August 2017 onwards. In total, 36 TSP and 31 PM$_1$ samples
were collected per filter type during the field campaign (including blanks).

### 2.2    Sample analysis

#### 2.2.1    Elements and water-soluble ions

The analysis of the elemental and water–soluble ion concentrations was performed at LISA
according to the protocols previously detailed in Klopper et al. (2020) and Desboeufs et al.
(2024). The elemental concentrations of 24 elements (Na, Mg, Al, Si, P, S, Cl, K, Ca, Ti, V, Cr,
Mn, Fe, Co, Ni, Cu, Zn, As, Sr, Pb, Nd, Cd, Ba) were measured by wavelength-dispersive X-
ray fluorescence (WD-XRF) using a PW-2404 spectrometer (Panalytical, Almelo,
Netherlands). The instrument was calibrated with mono- and bi-elemental certified elemental
standards (Micromatter Inc., Surrey, Canada). The concentrations of light-weight elements (Na
to Ca) in the TSP fraction were corrected for X-ray self-attenuation as described in Formenti
et al. (2010), assuming a mean diameter of 4.5 µm to represent the average coarse particle
size. Elements heavier than Ca, as well as concentrations measured in the PM$_1$ fraction, were
not corrected. The measured atmospheric concentrations are expressed in $ng\,m^{-3}$, and the
relative analytical uncertainty was evaluated as 10 %.
The analysis of the water-soluble fraction was performed by extracting the filters with 20 mL of
ultrapure water (MilliQ® 18.2 MΩ.cm) for 30 minutes. The solution was divided into two sub-
samples filtered to 0.2 µm of porosity (Nuclepore). One half was analyzed by Ion
chromatography (IC) using a Metrohm IC 850 device equipped with a column MetrosepA supp
7 (250/4.0 mm) for anions and with a Metrosep C4 (250/4.0 mm) for cations. The IC analysis
provided the concentrations of the following water-soluble ions: $F^-$, formate, acetate, $MSA^-$
(methanesulphonic acid), $Cl^-$, $NO_3^-$, $SO_4^{2-}$, oxalate, $Na^+$, $NH_4^+$, $K^+$, $Ca^{2+}$ and $Mg^{2+}$. A
calibration with certified standard multi-ions solutions of concentrations ranging from 5 to 5000
ppb was performed and, the uncertainty of the analysis was estimated to be 5%.
The second half of the solution was acidified to 1% with ultrapure nitric acid (HNO$_3$) and
analysed by a combination of inductively coupled plasma-atomic emission spectroscopy (ICP



AES) using Spectro ARCOS Ametek® ICP-AES and by high-resolution inductively coupled
plasma-mass spectrometry (HR-ICP-MS) using a Neptune Plus™ instrument by Thermo
Scientific™ as described in Desboeufs et al. (2024). The calibration curve was performed using
standard multi-element solutions ranging from 1 to 1000 ppt. The elemental fractional solubility
(FS) for element is calculated as the ratio between the dissolved and the total concentration.
Organic carbon (OC) and elemental carbon (EC) were measured using a thermo-optical
carbon analyser (Sunset Laboratory Inc.) on a 1.5 cm$^2$ filter following the EUSSAR-II protocol
(Cavalli et al., 2010). The Sunset analyzer was calibrated using a sucrose solution (purity >
99.5 %) in the concentration range between 0.42 μg cm$^{-2}$ and 40 μg cm$^{-2}$. The limit of
quantification for total carbon and organic carbon is henceforth estimated to be equal to 0.42
μg cm$^{-2}$. An instrumental blank and a control point with a sucrose solution at 10 μg cm$^{-2}$ were
done at the beginning of each day of analysis. OC and EC concentrations are automatically
calculated with the software OCBC835 (Sunset Laboratory). The optical split point was
manually verified to ensure their assignment.
All the concentration values presented in this paper were corrected for the average
concentration measured for their corresponding analytical blanks, which was almost equal to
the limit of detection.

### 2.2.2    Iron mineralogy

The quantification of iron oxides and the partitioning of iron species in the II- and III-oxidation
state was performed by X-Ray Absorption (XAS) analysis at the Fe K-edge. Analysis was
performed on the Teflon TSP filters only as the concentrations of the PM$_1$ filters were not high
enough for this kind of analysis.
XAS analysis was conducted at the SAMBA (Spectroscopies Applied to Materials based on
Absorption) line at the SOLEIL synchrotron facility in Saclay, France (Briois et al., 2011)
according to the protocols and procedures previously presented in Formenti et al. (2014) and
Caponi et al. (2017). A Si(220) double-crystal monochromator was used to produce a
monochromatic X-ray beam, which was 4000 x 1000 μm$^2$ in size at the focal point. The energy
of the X-ray beam was calibrated with an external Fe foil standard before the experiments. The
energy range was scanned from 7050 eV to 7350 eV at a step resolution of 0.2 eV.
Aerosol samples were mounted in an external setup. A portion of each aerosol filter sample
was cut and mounted on a carton board holder with 5 available positions and analysed in
fluorescence mode without prior preparation. The number of scans per sample was set
between 50 and 200, depending on the iron concentration, to improve the signal-to-noise ratio.
One scan acquisition lasted approximately 100 seconds for a total of 1.3 hours to 5.5 hours of
measurements for 50 to 200 scans.



The spectral analysis was conducted with the FASTOSH software package developed at
SAMBA. As described in Wilke et al. (2001) and O'Day et al. (2004), the oxidation state and
the bonding environment of Fe in dust samples give rise to different features in the XAS
spectra. In the pre-edge region, the shape of the XAS spectra is determined by electronic
transitions to empty bound states, which are strongly influenced by the oxidation state of the
absorbing atom but also by the local geometry around the absorbing atom due to hybridization
effects. Wilke et al. (2001) found that for Fe(II)-bearing minerals, the position of the centroid of
the pre-edge is found at 7112.1 eV, whereas it is at 7113.5 eV for Fe(III)-bearing minerals. The
position of the rising edge, which also depends on the oxidation state, is found at approximately
7120 eV. In the X-ray Absorption Near Edge Structure (XANES) region, extending
approximately 50 eV above the K-edge peak, features are determined by multiple-scattering
resonances of the photo-electron ejected at low kinetic energies.
The speciation of Fe was obtained by the least-square fit of the measured XANES spectra
based on the linear combination of mineralogical references. Fits were conducted on the first
derivative of the normalized spectral absorbance in the energy region between 7100 to 7180
eV, corresponding to -30 and +50 eV of the K-edge. Only the fits with a $\chi^2$ closest to 1 were
retained for further analysis.
The reference standards were chosen based on the expected iron mineralogy in the area
(White et al., 2007; Heine and Völkel, 2010; Formenti et al., 2014; Sracek, 2015; Zhang et al.,
2022). Standards for clays (illite and montomorrilonite) and iron oxides as goethite, magnetite
and hematite were taken from Formenti et al (2014) and Baldo et al. (2020). The ferrihydrite
standard was derived by the database of the Advanced Light Source, Lawrence Berkeley
National Lab (S. Frakra, pers. comm.). Standards for metal-ligand complexes expected to form
in fog droplets and deliquescent aerosol at high RH (Giorio et al., 2022) is provided in Table
S2 in the supplementary material.

### 2.2.3 Organic analysis

The water-soluble fraction of organic aerosols (WSOC) was extracted, purified using
hydrophobic resin (Bond Elut ppl) and analysed by high-resolution mass spectrometry (HRMS)
namely by hybrid LTQ-Orbitrap equipped with electrospray source (ESI) operated in negative
ion mode. All the samples were directly injected into the ESI source using a syringe pump. In
each case spectra were recorded in triplicates in two mass diapasons to decrease the number
of ions in the detector: 50-500 m/z and 150-700 m/z. Raw spectra were treated following the
laboratory procedure reported in Zherebker et al. (2024). Files were converted to *.mzML
format using msconvert with vendor-recommended peak-picking algorithm
(https://proteowizard.sourceforge.io/; last accessed 18/12/2024), which was suitable for a



series of in-house written Python scripts which included de-noising, calibration, formulae
assignment, and blank subtraction. Only formulae presented in triplicates were retained.
Diapasons were combined with removing lower-intensity duplicates. Formulae assignment
was performed considering only single-charged ions, ignoring anion-radicals with the following
atomic constraints: O/C ratio ≤ 2, 0.3 < H/C ratio < 2.5; element counts [1 < C ≤ 60, 2< H ≤
100, 0 < O ≤ 60, N ≤ 2, S ≤ 1]. Each formula was attributed to the tentative chemical class
(Table S4) based on the constrained aromaticity index ($AI_{con}$) calculated according to
Zherebker et al. (2022). In addition, the double bond equivalent (DBE) has been calculated,
which represents the sum of $sp^2$ and sp bonds and cycles. Further details of the mass-
spectrometry results are provided in Text S1 in the supplementary material.

### 2.3  Ancillary products

#### 2.3.1  Air mass back trajectories

Three-dimensional air mass back-trajectories ensemble are calculated using the NOAA HYbrid
Single-Particle Lagrangian Integrated Trajectory Model (HYSPLIT; Draxler and Rolph, 2015).
Weather Research Forecasting Model (WRF, version 3.71; Skamarock et al., 2008) forced by
Operational Global Analysis data (NCEP: National Centers for Environmental Prediction;
GDAS: Global Data Assimilation System, https://rda.ucar.edu/datasets/d083002, last
accessed 10/01/2025) was used to simulated hourly meteorological data at 50km and 9km
horizontal resolution.

#### 2.3.2  Atmospheric circulation

The atmospheric circulation and composition at the regional scale from 21 August to 13
September 2017 was further investigated using the Copernicus Atmospheric Monitoring
Service (CAMS) reanalysis (Inness et al., 2019), available every 3 h from 00:00 to 21:00 UTC.
The spatial resolution is approximately 80 km and 60 pressure levels (37 of which are below
20 km and 20 below 5 km). The atmospheric composition is described by analysing the total
aerosol optical depth (AOD) at 550 nm and the mass mixing ratio of dust and sulphate
aerosols. The atmospheric circulation is described by analysing the near-surface (10 m) wind,
to highlight emission processes and local transport, and the geopotential height at 700 hPa, to
highlight the large-scale circulation and long-range transport. Atmospheric composition and
circulation data are averaged at the daily time scale.

#### 2.3.3  Positive matrix factorisation analysis

Positive Matrix Factorisation (PMF) (Paatero, 1997; Paatero and Tapper, 1994) was applied
to chemical composition data (OC, EC, inorganic ions and total metals) of TSP and $PM_1$
samples using the software EPA PMF 5.0. Different factor solutions were investigated in the
range of 3 to 8 factors, starting from 10 different seeds. The 4-factor solution and the 3-factor



solution were selected for TSP and $PM_1$, respectively, based on the inflexion point of Q/Qexp
and chemical interpretation of the resulting factor profiles (loadings). The selected solutions
were run again from 100 different seeds, and the solutions with the lowest Q were selected.
Rotational ambiguity was investigated by changing the Fpeak parameter from 0 to ± 0.5 and ±
1. The solutions with Fpeak = 0 were selected, and bootstrap analysis was performed using a
number of bootstraps of 100 and a minimum r-value of 0.6.

## 3    Results

### 3.1    Air mass origin and local meteorology

Chazette et al. (2019) and Gaetani et al. (2021) showed that mid-tropospheric air masses
during the field campaign were characterized by three distinct periods. A first period (P1; 22–
28 August 2017) when air masses were southerly and characterized by low aerosol content
and large particles. From 23 to 25 August, the circulation in the middle troposphere was
characterized by the reinforcement of the South Atlantic anticyclone, leading to prevailing
south-westerly winds above Namibia (Figure S2). From 26 to 28 August, the transit of a
disturbance in the Southern Ocean was accompanied by the installation of the continental high
and prevailing north-westerly winds above Namibia (Figure S2). A second period (P2; 29
August–1 September 2017) when the circulation was characterized by the weakening of the
South Atlantic anticyclone and the reinforcement of the continental high (Figure S2),
associated with a northerly/easterly flow and transport of recirculation of a higher load of
aerosols associated with biomass burning. The circulation pattern remained the same on the
third period (P3; 3–12 September 2017), but the aerosol content further increased. After the
transit of a cut-off low in the upper troposphere on 2-4 September (Flamant et al., 2022), the
large-scale circulation was dominated by the further reinforcement of the continental high and,
on the 8-9 September, by the installation of a trough over the South Atlantic, leading to
favourable conditions for the recirculation of continental aerosol towards Namibia.
The same synoptic circulation was observed at the surface level. Air mass back trajectories
(Figure S3) show that the air flow at the surface level was southerly during P1 and P2 but
shifted to north-easterly (continental) after 2 September (P3), when the frequency of the anti-
cyclonic circulation towards Henties Bay increased. Continental air masses generally took
more than 2 days to reach the site. In the last two days of transport, they moved along the
coast or recirculated around Henties Bay, alternating the S-SW and NW-NNW directions. In a
few cases, and in particular on 11 September, the transport of continental air masses was
more direct and within two days from Henties Bay. The record of local winds measured during
the campaign (Figure S1 in the supplementary material) testifies of the frequent recirculation.
Strong winds (average 5 m s$^{-1}$, and up to 10 m s$^{-1}$) came, alternatively, from the S-SW direction



(20-22 August, 29-31 August, 6-9 September, 11 September, grey boxes in Figure S1) and
the NW-NNW direction (23-28 August, 1-5 September, 10 September, 12-13 September).
Occasionally, a gentle land breeze (easterly winds below 2 m s$^{-1}$) was observed before sunrise
or after sunset. In general terms, as discussed in Giorio et al. (2022), the local meteorological
conditions at Henties Bay during the campaign were characterized by remarkable stability in
terms of temperature (around 12 °C) and humidity (RH ~95%), while a persistent stratocumulus
cloud deck kept solar irradiance below 600 W m$^{-2}$.

### 3.2 Aerosol composition and origin

The summary statistics of the aerosol composition is reported in Table S3 as supplementary
information. The TSP chemical composition was dominated by the sea salt tracers, Na$^+$ and
Cl$^-$ (average ± standard deviation concentrations of 22 ± 11 µg m$^{-3}$ and 39 ± 21 µg m$^{-3}$,
respectively), as well as SO$_4^{2-}$ (9.1 ± 4.3 µg m$^{-3}$), Mg$^{2+}$ (3.9 ± 2.0 µg m$^{-3}$), K$^+$ (1.2 ± 0.6 µg m$^{-3}$)
and Ca$^{2+}$ (1.7 ± 0.8 µg m$^{-3}$). In terms of metals and metalloids, Al (0.6 ± 0.4 µg m$^{-3}$), Fe (0.6 ±
0.4 µg m$^{-3}$), and Si (2.5 ± 1.3 µg m$^{-3}$) had the highest concentrations. The mean OC and EC
concentrations were 3.2 ± 1.5 µg m$^{-3}$ and 0.2 ± 0.2 µg m$^{-3}$, respectively. The concentrations of
methanesulfonic acid (MSA), tracer of marine biogenic productivity, averaged at 61 ± 26 ng
m$^{-3}$. In the PM$_1$ fraction, due to the low flow rate used for sampling (10 L min$^{-1}$), only major
elements and ions were detected. Concentrations in the PM$_1$ fraction were generally lower
than in the TSP.

### 3.2.1 Marine aerosols

Figure 1 presents the time series of the elemental concentrations of Na$^+$ and Cl$^-$ and their ratios
(Cl$^-$/Na$^+$) in the TSP and PM$_1$ fractions.

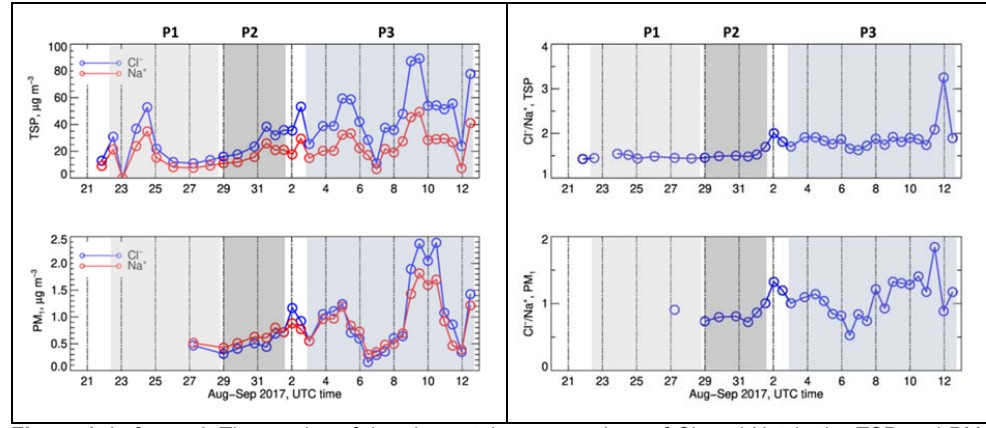

**Figure 1.** Left panel: Time series of the elemental concentrations of Cl$^-$ and Na$^+$ in the TSP and PM$_1$
fractions. Right panel: Time series of elemental ratio of Cl$^-$/Na$^+$ in the TSP and PM$_1$ fractions.




The concentrations of $Na^+$ and $Cl^-$, strongly correlated as expected, showed a relatively
constant background of approximately 10 µg m$^{-3}$ (TSP) and 0.3 µg m$^{-3}$ (PM$_1$), and intense
peaks of concentrations. In the TSP fraction, the $Cl^-$ concentration was up to 80 µg m$^{-3}$. The
$Cl^-/Na^+$ ratio was of the order of 1.5 in the P1 and P2 periods, and of the order of 1.8 afterwards.
In the PM$_1$ fraction, the $Cl^-$ concentration reached 2.5 µg m$^{-3}$. The $Cl^-/Na^+$ ratio, little
documented during P1, was around 0.7, while it increased between 1 and 1.8 during P2 and
P3. Values of the order of 1.5-1.8 are consistent with the composition of local seawater (Giorio
et al., this issue) and average sea spray (Seinfeld and Pandis, 2006), as well as the previous
results by Klopper et al. (2020).
The mass concentration of organic carbon (OC) through the campaign was strongly associated
with $Na^+$ as well as $Cl^-$ (not shown). The OC/$Na^+$ ratio was variable and ranged between 0.07
and 0.3, consistent with values reported by Frossard et al. (2014) for marine aerosol types.
The molecular analysis of the organic composition provides insights into the sources affecting
the OC/$Na^+$ ratio during the campaign (Figure 2).

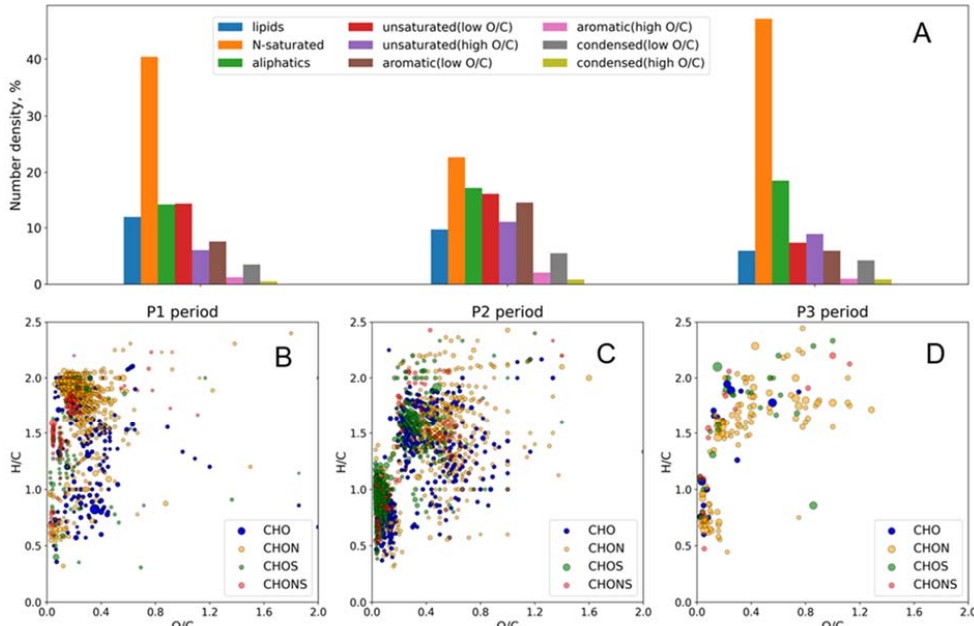


**Figure 2**. Population density of all molecular compositions based on Alcon classes (a), and van
Krevelen diagrams for only unique molecular assignments in samples under study in the three periods
(b-d), where unique formulae were determined only in a sample from the designated period. The size of
the points reflects relative intensities in the mass spectra (not used in the analysis).

The differences in the molecular composition of samples from the P1-P3 periods are depicted
in the van Krevelen diagrams (Figure 2b-d), which highlight the unique molecular composition



of each period. P1 was dominated by saturated and low oxidized (O/C < 0.3) CHO and CHON compounds, which occupy about 60% of the total molecular space (Figure 2). This may indicate the high contribution of biogenic fatty acids and protein-derived compounds (Bikkina et al., 2019). Their cumulative contribution in P2 decreased to about 40%, while a significant increase in the contribution of oxidized saturated compounds (O/C > 0.3) as well as highly unsaturated ($AI_{con}$ > 0.5) compounds was observed. Moreover, reduced (low O/C) saturated compounds appear to be unique for the P1 period (Figure 2b). This supports the biogenic source brought by south-westerly winds. Further, the contribution of continental dust with clear anthropogenic contribution is reflected as unique highly unsaturated compounds in the P2 period as well as an increase in S-containing compounds (Figure 2c). P3 was depleted with highly unsaturated compounds with a relative dominance of saturated N-containing compounds. The aerosol sources are similar between the P2 and P3 periods, which resulted in an insignificant amount of unique molecular assignment in the latter (Figure 2d). In addition, the double bond equivalent *vs.* molecular mass diagram in the supplementary material indicates an increase in the contribution of biomass-burning aerosols and possibly sulphate-enriched dust from smelting in the P2-P3 periods compared to the P1 period, which is in line with the air mass origin. Further details of the organic composition are reported in the supplementary materials.

### 3.2.2 Fluoride concentrations

The P1 and P2 periods were also characterised by extremely high concentrations of fluoride (up to 10 µg m$^{-3}$) as shown in Figure 3, in line with what reported by Klopper et al. (2020) for the PM$_{10}$ aerosols measured during 2016 and 2017 at the site. In September (P3), the concentration of F$^-$ dropped to zero, as a consequence of the change in the origin of the air masses transported to the site.

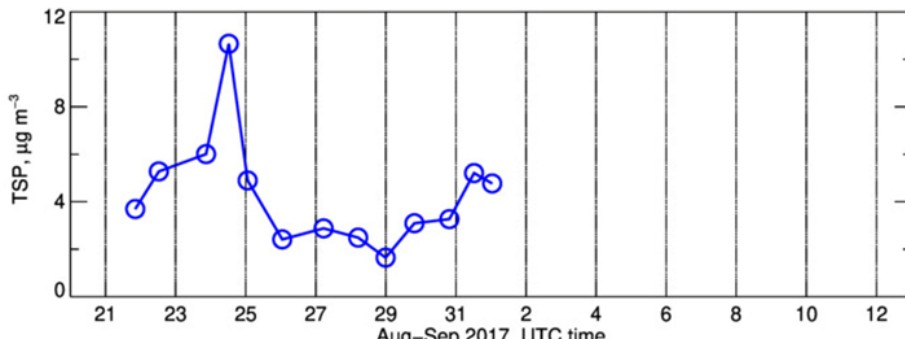

**Figure 3.** Time series of the elemental concentrations of F$^-$ in the TSP fraction during the field campaign.



Fluoride is a natural occurring ion in marine environments as well as in mineral dust (Fuge, 2019). In Namibia, the release of dissolved fluoride to the atmosphere is due to the evaporation of fluoride-rich groundwater (Sracek et al., 2015) or the erosion of mineral deposits of calcium fluoride ($CaF_2$, Onipe et al., 2020). Fluoride is present in significant amounts (> 1 wt.% F) in francolite, a carbonate fluorapatite mineral (typical formula $Ca_{4.7}Na_{0.2}Mg_{0.1}(PO_4)_{2.6}(CO_3)_{0.4}F_{1.28}$), which can be found in phosphorite deposits on the Namibian shelf, notably in the area between 23° and 25.5°S south of Henties Bay (Compton and Bergh, 2016; Mänd et al., 2018). This is likely to be the origin of the excessive fluoride concentrations observed during the P1 period of the campaign. Not only the origin of air masses detected at Henties Bay coincided with the locations of the marine deposits, but during P1 the fluoride content correlated with major marine tracers (Na, Cl, S), and with calcium, both its sea salt and non-sea-salt fractions (nss-$Ca^{2+}$/$F^-$ ratio ranging from 0.1 to 0.3, as in Klopper et al. (2020)), as well as with P, K and Sr, the latter can replace Ca in the francolite mineral structure (Compton and Bergh, 2016; Rakovan and Hughes, 2000).

### 3.2.3 Mineral dust composition

Figure 4 presents the time series of the elemental concentrations of Al, Si and Fe, major tracers of mineral dust, as well as of their ratios (Si/Al, Fe/Al and Fe/Si) in the TSP and $PM_1$ fractions.

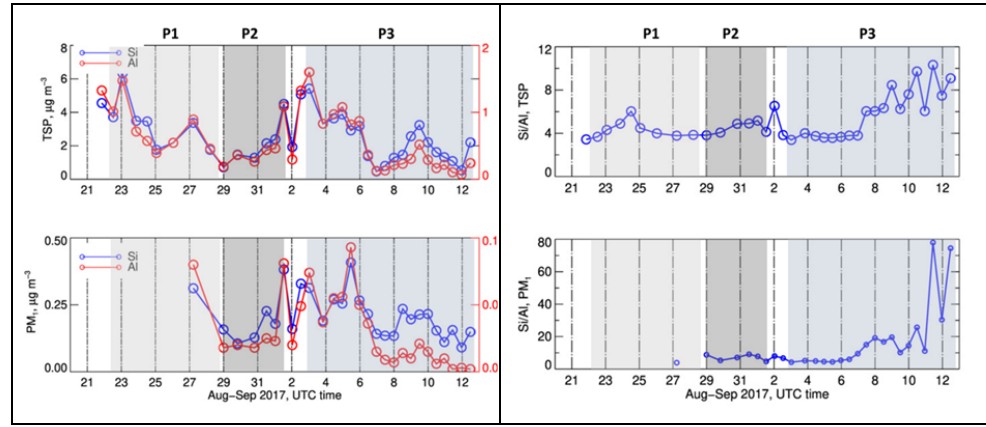

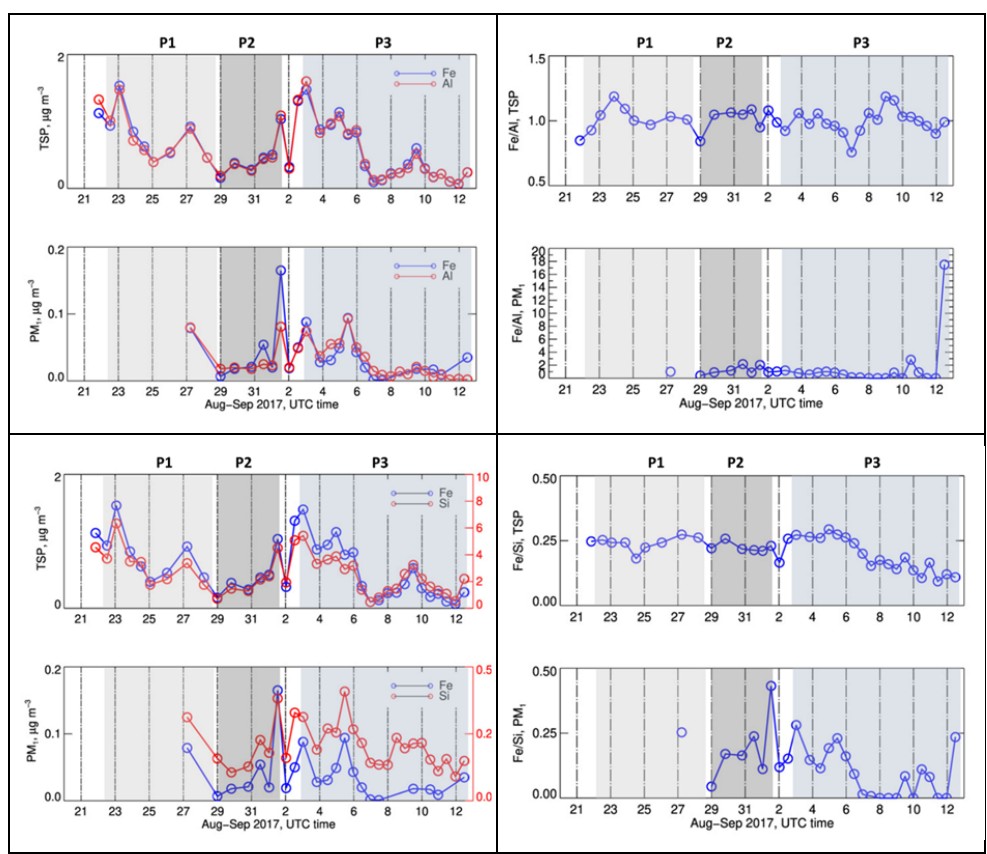

**Figure 4.** Left panel, from top to bottom: Time series of the elemental concentrations Al, SI and Fe in
the TSP and PM$_1$ fractions. Right panel from top to bottom: Time series of elemental ratio of Si/Al, Fe/Al
and Fe/Si in TSP and PM$_1$ fractions.

The elemental concentrations of Si and Al were up to 6 and 1 µg m$^{-3}$ in the TSP fraction and
up to 410 and 90 ng m$^{-3}$ in the PM$_1$ fraction, respectively. In the P1 and P2 periods, and with
the exception of a peak value on 26 August, the Si/Al ratio was of the order of 3.8, consistent
with the findings of Klopper et al. (2020) for natural mineral dust emitted from the Namibian
gravel planes. After this date, that is during P3, the Si/Al ratio increased to values between 6
and 10 (TSP) and between 8 and 80 (PM$_1$), indicating a very strong enrichment with respect
to the composition of the regional mineral dust.
In the TSP fraction, and regardless of the period, the Fe/Al ratio was in the range of 0.8-1.2,
as previously found for the natural gravel plain dust in the area (Eltayeb et al., 1993; Klopper
et al., 2020). Likewise, during P1 and P2 the Fe/Si ratio was consistent with those previous
observations for mineral dust, and so was the Fe/Ca ratio (not shown), found in the range 0.2-
0.8. However, during P3, the Fe/Si decreased from approximately 0.25 to 0.1, while the Fe/Ca



ratio increased to between 0.09-0.15 (not shown). In the PM$_1$ fraction, both the Fe/Al and the
Fe/Si ratios were more variable with time (only the September period is documented). The
Fe/Al ratio was of the order of 1 as in the TSP fraction, except on the last day of the campaign
when the ratio reached 18. The Fe/Si was higher between 29 August and 6 September (e.g.,
the P2 period), ranging from 0.2 to 0.4, and decreased to 0.1-0.2 in the last days of the
campaign. That corresponded to the variability of several major elements and metals (K, Mg,
Co, Cu, Nd, Ni, Sr, Cd, but Zn, As and Pb in particular), whose ratio with Al were significantly
higher during the last days of the campaign (7-12 September, Figure S5). As for OC/Na$^+$, the
aerosol mineral composition of TSP during the P3 period could be split into two sub-periods
before and after 6 September, with an enrichment in metals and OC at the end of the campaign.

### 3.2.4   Source apportionment

These observations are reflected by the PMF analysis, described in Text S2 in the
supplementary material. Note that, despite its high concentrations, fluoride was not included
in the source apportionment because it was not measured during the whole field campaign. In
the TSP fraction, the analysis separates two factors characterized by a high loading of metals.
The first one is a "mineral dust" factor characterized by Al, Fe, and Si, as well as Ti, Mn, Na$^+$,
Ca$^{2+}$, and SO$_4^{2-}$. Its contribution, significant only during the first part of the campaign,
accounted, on average, for 5.8% of total TSP mass. The second factor, called "Si-rich", is
characterized by the presence of a high loading of Si, As and Pb, strongly correlated to each
other (r=0.97), moderate loadings of Co, Cu, Ni, Nd, Sr, Zn and EC, but not correlated to Al
nor Fe, contrarily to "mineral dust". The "Si-rich" factor, significant mostly during the P3 period,
notably after the 6 September, accounted for 23.3% of TSP mass but was not found in the PM$_1$
fraction where the concentrations of the majority of its tracers were very close to the detection
limit. This is in agreement with the fact that the fraction of coarse particles with respect to the
total number increased on the last days of the campaign (Figure S6 in the supplementary
material).
The chemical fingerprinting of the "Si-rich" factor is similar to that reported from windblown dust
from mines in the Otavi Mountainland in Namibia (Mileusnić et al., 2014; Sracek, 2015) as in
the Zambian Copper Belt (Meter et al., 1999; Ettler et al., 2011; 2014; Mwaanga et al., 2019),
a large and important mining area in the northern part of Zambia (Aurélien et al., 2022;
Martinez-Alonso et al., 2023; Křížek et al., 2023). Sracek (2015) found various associations of
Fe with Cu, Co, Pb, V, As, Pb, and Zn for mines in Zambia and Namibia, characterized by
different climates and ages of the core. At a receptor site on the Zambian Copperbelt in
Zimbabwe, the analysis by Nyanganyura et al. (2007) identified the mixing of mineral dust and
metal smelting emissions by distinguishing the long-range transport of a aerosols containing
Fe, Al, Si but also Co from a non-ferrous smelter component contributing to the fine aerosol



fraction only, and characterized by S, Zn, As, and Pb. Ettler et al. (2007) indicated that iron is
enriched both with respect to Al and Si in dust liberated from Cu–Co metal smelters in the
Zambian Copperbelt. However, Meter et al. (1999) showed that enrichment is variable on an
event basis depending on the pyro-metallurgical processing of ores and their composition.
We henceforth conclude that during P1 and P2 the aerosol composition was dominated by
natural Namibian mineral dust sources with a composition very similar to the average dust
composition measured at the same site in 2016-2017 (Klopper et al., 2020) and reaching the
site within the southerly air flow. Conversely, during the last part of P3, the dust reaching
Henties Bay was fugitive material from anthropogenic activities. The air masses during the
second part of P3 could originate as far as from the Zambian Copper belt.
These conclusions are further corroborated by the CAMS reanalysis shown in Figure S7 in the
supplementary material. During P1 and P2 the dust mass mixing ratio reached up to 120 µg
m$^{-3}$ (100 µg kg$^{-1}$ on the CAMS map) at the surface in correspondence with the coastal sources
in Namibia as a response to the prevailing south-easterly winds (Figure S4) dominating the
near-surface circulation from 23 August to 3 September. Continental dust sources were
activated on 22 August and 1 September, in association with south-westerly near-surface
winds, and on 28-29 August and 4-5 September, in association with near surface convergence
of south-westerly and north-easterly winds. From 6 September onwards, no remarkable dust
activity was observed, while the circulation had changed (Figure S7). The CAMS reanalysis
also showed that the sulphate mixing ratios at the surface reached 6 µg m$^{-3}$ (5 µg kg$^{-1}$ on the
CAMS map) in the Zambian Copper Belt and in the urban area of Pretoria and Johannesburg
(Figure S7), also a known pollution hotspot (Martinez-Alonso et al., 2023). Sulphate aerosols
remained close to their source regions until the end P2. With the installation of the continental
high on the 3 September, sulphate aerosols were recirculated south-westwards towards
Henties Bay during P3, in particular during 10-12 September.
**3.3    Iron mineralogy**
The first derivative of the four XANES normalized spectra corresponding to the highest Fe
concentrations measured during AEROCLO-sA are shown in Figure 5. The remaining spectra,
including those of the standard minerals and compounds used for the deconvolution, are
reported in Figure S8.



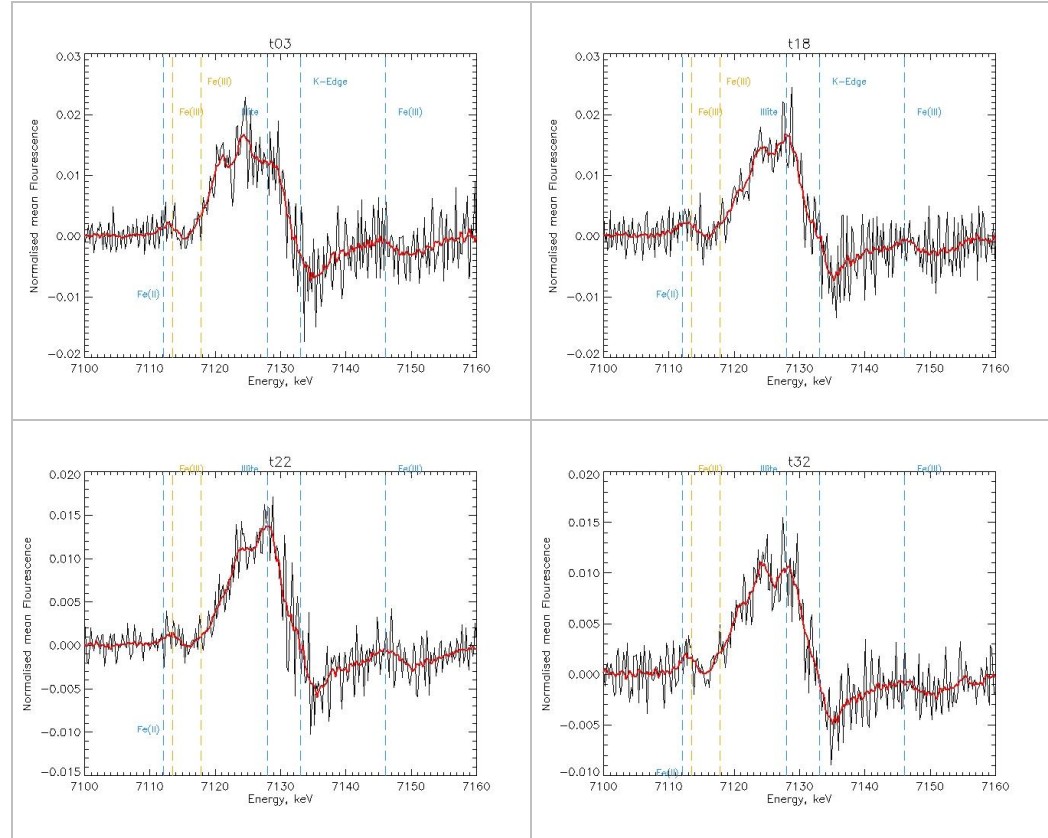

**Figure 5**. First derivative of the four XANES normalized spectra corresponding to the highest Fe concentrations measured during AEROCLO-sA (see Table S3). The spectral positions of the absorption bands of Fe(II) and Fe(III) in the pre-edge region (7112.1 to 7117.8 eV) as well as those of illite (7128 eV) and various Fe(III) minerals, including hematite (7146 eV), according to Wilke et al. (2001) are indicated by vertical intermittent lines.

Because of the small quantities of particle mass collected on the filters, the XANES spectra are rather noisy. The main features can nevertheless be explored after smoothing. They all are rather similar. In the region between 7122 and 7128 eV, two to three peaks are present with different intensities depending on the sample. The peak at 7128 eV is minor on sample T03 (P1 period) for which the peaks at 7122 and 7126 eV dominate. The peak at 7122 eV is not present afterwards. For samples T18 and T22 in the P2 and beginning of P3 periods, only the peaks at 7126 and 7128 eV are present, the intensity of the latter being higher than the one of the former, while those peaks have equal intensity on sample T032 in the P3 period. For mineral dust from northern Africa, Formenti et al. (2014) showed that the relative proportions of these peaks can be related to the type of clays, but also to the presence of iron oxides in the form of hematite ($Fe_2O_3$) or goethite (FeOOH). Peaks between 7132 and 7136 eV are distinctive of clays and iron oxides in the form of hematite, but are not present for





goethite. In our samples, a minor shoulder in this spectral region is observed only for a few
samples (T24, T25, and T31) collected in September. On the other hand, the pre-edge region
between 7110 and 7116 eV is sensitive to the iron oxidation state. The majority of our samples
seem to peak around 7113-7114 eV, indicating that iron is predominantly in the Fe(III) oxidation
state. Only for a few samples (T10, T25, T26, T30, and T37), the pre-edge peak is closer to
7112 eV indicating that Fe(II) could be the predominant oxidation state.
Several attempts of least square reduction were done with a variable number of references to
reflect the many mineralogical forms in which iron can be found and verify the stability of the
solution. While the relative proportions might have changed by a few percent, the overall
repartition was found to be consistent and independent of the selected references.
The average least-square apportionment of the total TSP elemental iron is presented in Table
1 in terms of the mean fractions par sampling period and mineralogical classes.

**Table 1.** Apportionment of total iron (percent mean and standard deviation) by the least-square
deconvolution of the XANES spectra obtained on the filter samples indicated in the first row. Results are
grouped by period and by mineral classes.

|  | 21-31 Aug 2017 T01-T14 | 1-2 Sep 2017 T15-T18 | 2-7 Sep 2017 T20-T27 | 8-12 Sep 2017 T28-T38 |
|---|---|---|---|---|
| **Clays** | 49 ± 13 | 58 ± 12 | 53 ± 11 | 42 ± 10 |
| **Iron oxides** | 40 ± 10 | 36 ± 8 | 40 ± 9 | 46 ± 8 |
| **Oxalate** | 4 ± 4 | 3 ± 4 | 3 ± 4 | 5 ± 7 |
| **Pyrite** | 7 ± 5 | 3 ± 4 | 4 ± 5 | 7 ± 4 |


The largest contribution to the total iron is by clays, between 42 to 58%, with illite and
montmorillonite contributing in equal proportions. While a clear temporal trend cannot be
defined, the contribution of clays is lowest during the latest sampling period (6-12 September
2017). The second largest contribution is by iron oxides, accounting for between 38 to 45% of
the total iron throughout the sampling period. The contribution of FeO, iron oxide in Fe(II) form,
is low and extremely variable from sample to sample and not necessarily retrieved for samples
in which a shift towards the Fe(II) seems evident in the pre-edge region (T10, T25, T26, T30,
and T37). The period P3 is also characterized by the lowest contribution of Fe(III) oxide (64%
vs 72-75%). Ferrihydrite (14 ± 7% of total iron) and goethite (8 ± 6% of total iron) showed their
highest contributions during P2 (21% and 12% respectively), while hematite (8 ± 6 of total iron)
was highest in September (P3). Sracek (2015) found that the formation of secondary hematite
is favoured by tropical climate conditions in mines in Zambia compared to Namibia. In contrast
to northern African dust, the least-square reduction shows that the presence of magnetite is
significant in the dust collected during the campaign, contributing 10-15% to the total iron oxide
fraction. Magnetite can be found in sediments in the Erongo region of coastal Namibia



(Lohmeier et al. 2021), but also in anthropogenic emissions, such as for example those from
metal smelting (Rathod et al. 2020). Zhang et al. (2022) investigated the light-absorbing
properties and single particle composition from airborne measurements offshore central Africa
during the ORACLES 2018 campaign, and attributed the presence of magnetite to the high-
temperature conversion of hematite and/or goethite in biomass-burning plumes or to industrial
or vehicular emissions, including from pyro-metallurgical processing in the Zambian
Copperbelt. Pyrite (FeS) and Fe-oxalate complexes were also detected throughout the
campaign, by their average contribution was extremely low, with the exceptions of 28-29
August 2017 (T10, approximately 19%) and 11 September (T35-36, approximately 30%).
**3.4  Iron solubility**
Figure 6 presents the time series of the fractional solubility for Al, Si and Fe, as well as those
of fluoride (F⁻) and MSA measured in the TSP fraction during the field campaign.

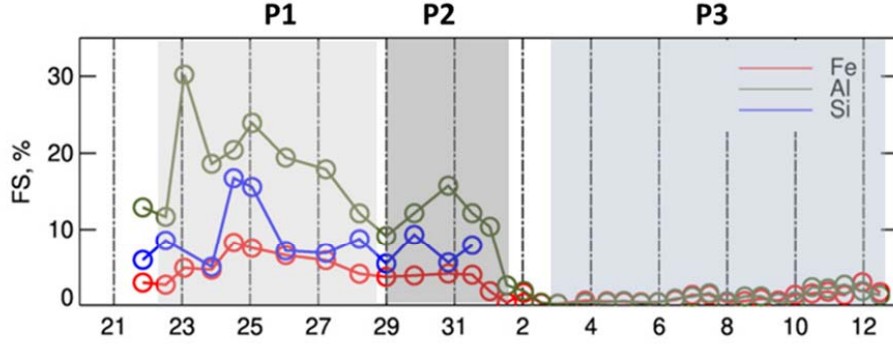

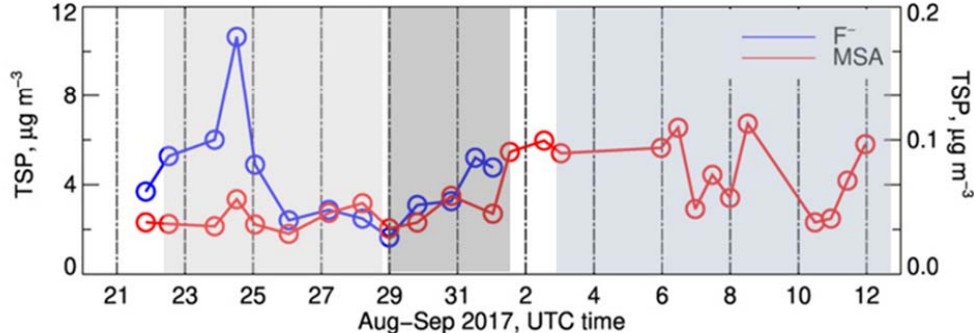


**Figure 6.** Top panel: Time series of the fractional solubility Al, Si and Fe in the TSP fraction. Bottom
panel: Time series of concentrations of F⁻ (blue) and MSA (red) in the TSP fraction. The MSA
concentrations are reported on the right axis of lower panel.



During P1 and P2, the fractional solubility was measurable and of the order of 2-8% for Fe, 9-
24% for Al, 5-17% for Si and 1.5-4% for Ti (not shown). After that, during P3, their fractional
solubility drastically dropped to 0.2-1.7% for Fe and 0.2-2.6% for Al, whereas the fractional
solubility of Si and Ti was not measurable (dissolved concentrations under limit of detection).
A similar behaviour was observed for most of the measured trace metals (As, Co, Cr, Cu, Ni,
Pb, Ti, and Zn, Figure S9).
The percent fractional solubility of iron measured during P1 and P2 was of the same order of
magnitude as the lowest solubility values reported at the same sampling site for $PM_{10}$ dust
particles by Desboeufs et al. (2024) for the period April to December 2017, when values as
high as 20% were measured when MSA in the particle phase was most concentrated. These
authors attributed the enhanced solubility to processing (photo-reduction) of the dust by gas-
phase dimethyl sulphide (DMS) emitted by the coastal Benguela upwelling. In the present
dataset, this association cannot be made. Figure 6 shows that the MSA concentrations were
of the same order of magnitude throughout the campaign, and actually slightly more
concentrated during P2 and P3, when, on the other hand, the Fe fractional solubility was the
lowest. Soluble Fe was also not correlated with oxalate (Pearson correlation coefficient r ~
0.3), another renown organic ligands (Paris and Desboeufs, 2013).
On the other hand, Figure 6 shows that during P1 and P2, the temporal variability of the
fractional solubility of Al, Si and Fe closely followed that of the mass concentration of fluoride
(correlation coefficient of 0.87, 0.85 and 0.81, respectively). Fluoride has been identified to be
a good ligand of metals in aqueous solution, notably Fe(III) in comparison to Fe(II) (Connick et
al., 1956; Bond and Hefter, 1980). The abundance of fluoride ions could act to facilitate the
metal complexation on the particle surfaces, potentially promoting their release from the bulk
oxide and dissolution at the solid/liquid interface (Arnesen et al., 1998; Tao et al., 2022). In
particular, $F^-$ may contribute to the disruption of the Si-O lattice bond by forming $SiF_6^{2-}$
complexes at acidic pH (Mitra and Rimstidt, 2009).

## 583 4 Conclusive remarks

The three weeks of aerosol sampling at the Henties Bay coastal site in Namibia during the
AEROCLO-sA field campaign coincided with a transition period between two synoptic regimes:
the dominance of southerly air flow, associated with the reinforcement of the South Atlantic
anticyclone (22 to 31 August 2017) and the dominance of north-easterly air flow (1 to 12
September 2017), associated with the installation of the mid-tropospheric continental high.
Those synoptic regimes corresponded to a significant change in the aerosol composition
measured at the site and in particular of that of mineral dust. During August and the first few
days of September, the dust was natural windblown from the southerly gravel plains with a



composition consistent with that previously found in Namibia (Klopper et al., 2020). Gater, the
dust was fugitive from anthropogenic mining and possibly also from smelting emissions as far
as in the Zambian Copper Belt. The anthropogenic influence in the latter part of the campaign
was also documented by the composition of the organic aerosol, which was rich in highly
unsaturated compounds as well as saturated N-containing compounds in the latter two
periods, more typical of anthropogenic pollution. A second major difference in the composition
of the air masses was the high fluoride content until September 2 attributed to emissions from
the marine shelf south of Henties Bay.
Taking advantage of those differences, this paper presents the first case study analysis of
differences and similarities in the composition of natural and anthropogenic dust, with two key
findings: (1) the elemental composition of the anthropogenic dust is enriched in silicon and
heavy metals, notably As, Mn, Cu, Cd, Pb, and Zn, and depleted in Al; (2) metals in
anthropogenic dust are less water-soluble than in the natural aeolian dust. In particular, the
fractional solubility of iron in the natural dust ranged between 2 and 8%, but remained lower
than 2% in the anthropogenic dust. This is rather unexpected when taking into account the
current literature on anthropogenic dust influenced by combustion, reporting that the iron
solubility would be the order of 50% (e.g., Li et al., 2017; Ueda et al., 2023). There are various
possible explanations to this fact. First of all, the mineralogy of iron. The most soluble form,
ferrihydrite (Journet et al., 2008; Shi et al., 2012), was more abundant in the natural dust, while
the less soluble forms of iron (iron oxides such as hematite and magnetite) were more frequent
in the fugitive dust, which conversely, could be more efficient in absorbing light at short
wavelengths. Secondly, during the first part of the campaign, the aerosol particles were smaller
in size, which is known to promote particle solubility, both directly but also in an indirect way,
allowing more intense atmospheric processing (Hamilton et al., 2022). Thirdly, our results
indicate in a very clear way the extent of which the solubility of iron is linked to the abundance
of fluoride ions during the first part of the campaign. While we do not have insights in the
mineralogical forms of the metals other than iron, the similar behaviour of their dissolved
concentrations, in particular Al and Si, suggest that the marine emissions of fluoride from the
Benguela shelf could play a key role in sustaining the complexation of metals dust particles
and facilitate their dissolution, supplementing the processing by DMS described for iron in
Desboeufs et al. (2024). Such high concentrations of $F^-$ ions are not only unexpected but also
they open questions for further studies in this environment. Similarly, to what is known for
chloride or bromine (Finlayson-Pitts, 2010, Simpson et al., 2015) one cannot exclude recycling
$F^-$ into reactive fluorinated radicals through heterogeneous processes. This call for further
targeted reanalysis of the organic matter sampled during this campaign both the gaseous and
particulate phases and for further laboratory work to investigate this quite poorly know



chemistry. Finally, our results suggest that, in the absence of processing by DMS or oceanic
fluoride, the transport of mining dust, including from the Zambian Copper Belt, is unlikely to be
a significant source of dissolved iron, but also of elements such as Mn, Cu and Zn, which are
toxic to phytoplankton even at low concentrations, and if assimilated, could alter the oceanic
productivity and microbial biogeochemistry (Adriano, 2001; Jordi et al., 2012; Mahowald et al.,
2018; Yang et al. 2019).
Future work should expand these results by addressing the frequency and intensity of those
occurrences on a longer time scale, as well as the mineralogy of metals and their processing
by marine emissions in the laboratory.



**Data Availability.** All data are made freely available by the French national service for atmospheric data AERIS-SEDOO data at https://baobab.sedoo.fr/AEROCLO/.

**Code Availability.** The FASTOSH XANES data analysis package is available for download at https://www.synchrotron-soleil.fr/fr/lignes-de-lumiere/samba (last accessed: 02/03/2024).

**Author contribution.** PF coordinated the AEROCLO-sA project and funding, led the field campaign and the data analysis, and wrote the manuscript with contributions from all the co-authors. SM performed PMF analysis. AZ performed ESI-HRMS analysis and analysed data. CG collected samples, supervised PMF and ESI-HRMS analyses and their data interpretation. CB contributed to the interpretation of the dust solubility and composition. KD analysed the fractional solubility measurements. MG analysed CAMS reanalysis. GS performed back-trajectories calculations. SC performed XRF analysis, ST performed IC / ICP analysis, FB and CDB performed XANES measurements under the supervision of GL. AF, JFD, AN and SJP participated and facilitated the field campaign.

**Special issue statement.** This article is part of the special issue "New observations and related modelling studies of the aerosol–cloud–climate system in the Southeast Atlantic and southern Africa regions (ACP/AMT inter-journal SI)". It is not associated with a conference.

**Competing interests.** Some authors are members of the editorial board of journal ACP.

**Acknowledgements**

Authors are grateful to the AEROCLO-sA consortium for their work in the field and during the preparation of the field campaign, and SANUMARC for hosting the field campaign. The authors wish to thank AERIS (https://www.aeris-data.fr/), the French center for atmospheric data and service, for providing the campaign website and organizing the curation and open distribution of AEROCLO-sA data.

**Funding support**

This work was supported by the French National Research Agency under grant agreement n° ANR-15-CE01-0014-01, the French national program LEFE/INSU, the French National Agency for Space Studies (CNES), and the South African National Research Foundation (NRF) under grant UID 105958. CG's work was supported by the Supporting TAlent in ReSearch@University of Padova STARS-StG "MOCAA", and a BP Next Generation fellowship awarded by the Yusuf Hamied Department of Chemistry at the University of Cambridge. This research received additional resources through the "The role of Secondary Organic Aerosols on the climate over the west coast of southern Africa (SOA-Clim)", International Research Project supported by University of Cambridge and CNRS. MG was supported by the project "Dipartimento di Eccellenza 2023–2027", funded by the Italian Ministry of Education, University and Research at IUSS Pavia. The PEGASUS facility receives funding as a national facility (instrument national) of the CNRS INSU.



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
