# Peer review of "Part 2: Elemental composition, iron mineralogy and solubility of anthropogenic and natural mineral dust aerosols in Namibia: a case study analysis from the AEROCLO-sA campaign"

_EGUsphere, 2025_

## Author Comment (AC1)

**Reply to referee comments (RC) to manuscript egusphere-2025-446**

The authors are grateful to Referee#2 for providing very valuable feedbacks to the manuscript. Our detailed point-by-point response to their comments are reported below.

We believe that the modifications requested by the reviewers improved the reading and the understanding of the manuscript, fostering its significance.

**General comments**

Model predictions of metal speciation and solubility in aerosols are highly uncertain, especially in Namibia. The authors collected PM1 and total suspended particulate (TSP) samples at Henties Bay during the period from 21 August to 13 September 2017. They combined wavelength-dispersive X ray fluorescence (WD-XRF), Ion chromatography (IC), X-Ray Absorption (XAS), Inductively Coupled Plasma Mass Spectrometry (ICP-MS), a thermo-optical carbon analyser, and high-resolution mass spectrometry (HRMS) to investigate the iron mineralogy and solubility. They found two major sources of iron which are influenced by natural and anthropogenic dust. Previous study at the Henties Bay Aerosol Observatory (HBAO) showed strong link between MSA (methane sulfonic acid) and dust iron solubility in PM10 up to 20%. Conversely, their results showed that dust iron solubility was correlated to the high concentrations of fluoride ion in TSP. They hypothesize that the source of fluoride ion is attributed to marine emissions from the Namibian shelf. The comprehensive measurements performed in this paper may help us to advance our understanding of metal speciation and solubility in aerosols, although more work is needed to confirm their hypothesis. I have some comments and questions to improve this paper.

**Specific comments**

l.41-44: Please consider separating one sentence to two sentences to explain the differences in solubility between the natural and anthropogenic dust aerosols. As you mention, a clear temporal trend of the higher content of iron oxides cannot be defined from Table 1. Figure S6 shows the coarse particle number concentration. Please show the higher content of iron oxides (see comment on l.611) and the larger size of particles in the anthropogenic dust (see comment on l.447).

l.53: Please describe the global dust emissions by anthropogenic activities quantitatively and add the reference.

This is now done

l.333: It is helpful for the reader if you refer to MQL in Table S3. If not, please specify major elements and ions.

This is now done in Table S3.

l.351: Please indicate PM1. If not, please specify the size of OC and NaCl.

This is now done

l.389 and Figure 3: Please show fluoride and MSA in PM1.

This is now done

l.401: Please show the figure with statistics for the correlation.

Correlation coefficients are mentioned in the text

l.326 and l.438: It is helpful for the reader if you show the figure of source apportionment. Please mention that the PMF analysis captured the major factor of TSP at first, possibly the sea salt.

This is now done. Figure 1 showing the results of the PMF analysis both in the TSP and PM1 fractions is now added to the main text and accompanied by a text of explanation (lines 344-363)

l.447 and Figure S6: Please show the fraction of coarse particles with respect to the total number.

Please see below

[Figure]

l.477: It is helpful for the reader if you show the figure of source apportionment from the CAMS reanalysis at the site in association with the PMF analysis.

The following figure shows the time series of the percent composition seen by CAMS at the Henties Bay site. While there is some similarity with the in situ measurements, a PMF analysis is not relevant because of the low time resolution and the limited number of compounds of CAMS.

[Figure]

l.481: Please show the comparison of sulfate between measurements and the CAMS reanalysis at the site.

The figure here below illustrates the comparison between the surface concentrations of sulphate measured at the site as reported for 2016 and 2017 by Klopper et al. (2020) and the surface measurements (top panel, blue line) and AOD (low panel, blue line) by CAMS. There is some degree of co-variance, the values of surface concentrations reported by CAMS are lower than measured at the station, possibly due to the fact that at the CAMS reported values refer to non-sea-salt sulfate only while measurements at the station are for total sulfate. This investigation is beyond the scope of this paper but could be pursued in the future.

[Figure]

l.532: You mention the higher content of iron oxides in the anthropogenic dust on l.42. Please specify Fe(III) oxide and show the results before grouping in supplement. How could you tell Fe(III) oxide from magnetite?

The identification of magnetite with respect to other Fe(III) oxides is based on the use of standards whose spectral signature is very different (see Figure S9 in the supplementary material).

l.610: Please show the results of the more abundant ferrihydrite in the natural dust during P1.
l.611: Please show the results of the more frequent iron oxides in the fugitive dust.

The percent fractions of ferrihydrite, hematite, goethite and hematite are now shown in Table 1.

l.613: Please show higher solubility in PM1 than TSP.

Unfortunately, the detection limit was too high and no analysis of the solubility of the PM1 could be performed

l.619: How do you consider the mechanism of fluoride emission from a carbonate fluorapatite mineral in phosphorite deposits on the Namibian shelf?

Atlas and Pytkowicz (1977) and Hossein et al (2024) describe the mechanism by which F could be released in sea water by dissolution. Upon dissolution, the release of F− to the atmosphere

can be attributed to the reaction with hydrogen in water to form hydrogen fluoride gas (or a solution of hydrofluoric acid; Anbar, M., and Neta, 1967). The high content of fluoride in the Namibian soil is also documented and attributed to weathering and dissolution of fluoride-containing minerals (Hossein et al, 2024). These comments and references are now added to the manuscript.

l.620: How do you consider the mechanism of mixing of fluoride and mineral dust?

In Namibia, the release of dissolved fluoride to the atmosphere is due to the evaporation of fluoride-rich groundwater (Sracek et al., 2015) or the erosion of mineral deposits of calcium fluoride (CaF2, Onipe et al., 2020). The mixing could also result from complexation

l.621: Please discuss the reasons of zero fluoride concentrations from marine emissions even though the MSA concentrations were higher during P2 and P3 than P1. How could you explain the lower solubility during P2 and P3 than P1 by supplementing the processing by DMS described for iron?

We have now added the times series of MSA and F in Figure 4, and a comment about MSA concentrations. As the site is coastal, there is always some contribution from marine emissions, which justify the MSA concentrations, which remained measurable but low (< 0.1 µg m-3) and displaying little variability. The F concentrations, if really attributable to marine emission, are coming from a specific area and specific air flow direction. We believe that there is no contradiction between the two possible mechanisms.

Technical comments

l.65, 72, and 74: Please add the references.

We thank the referee for carefully check the manuscript but all the statements at these lines have references already.

l.592: Please correct a typo.

This is now done

References

Anbar, M., and Neta, P.: Reaction of fluoride ions with hydrogen atoms in aqueous solution, Transactions of the Faraday Society, 63, 141-146, 10.1039/tf9676300141, 1967.

Atlas, E., and Pytkowicz, R. M.: Solubility behavior of apatites in seawater, Limnology and Oceanography, 22, 290-300, https://doi.org/10.4319/lo.1977.22.2.0290, 1977.

Hossein, M., Rwiza, M. J., Nyanza, E. C., Bakari, R., Ripanda, A., Nkrumah, S., Selemani, J. R., and Machunda, R. L.: Fluoride contamination a silent global water crisis: A Case of Africa, Scientific African, 26, e02485, https://doi.org/10.1016/j.sciaf.2024.e02485, 2024.

---

## Author Comment (AC2)

**Reply to referee comments (RC) to manuscript egusphere- 2025-446**

The authors are grateful to Referee#1 for providing very valuable feedbacks to the manuscript. Our detailed point-by-point response to their comments are reported below.

We believe that the modifications requested by the reviewers improved the reading and the understanding of the manuscript, fostering its significance.

**Review of "Elemental composition, iron mineralogy and solubility of anthropogenic and natural mineral dust aerosols in Namibia: a case study analysis from the AEROCLO-sA campaign" by Formenti et al.**

This manuscript presents the results of aerosol measurements conducted at Henties Bay, Namibia, with a focus on ionic and elemental composition in total suspended particles. Two regimes were identified, one related to regional dust and the other related to dust from anthropogenic activities. The paper is well written and provides insightful results to a region underrepresented in the literature.

General Comments:

- Much of the discussion of the results relies on the relationship between different elements (i.e. Figure 4), however, it is difficult to tell by eye when a change in the ratio is significant. The authors might consider placing error bars on the time series data, especially on the plots showing the time series of the elemental ratios.

    All figures with time series of concentrations and ratios have been modified to include the error bars.

- More discussion in the differences between the $PM_1$ and TSP composition would be useful. Currently, the $PM_1$ results are described (i.e. lines 356-350, throughout section 3.2.3) but additional insight into what the authors think is causing these differences would strengthen the paper. Additionally, it is unclear whether PMF was run on the TSP samples or both, as there is only one sentence (line 189-191) alluding to the $PM_1$ PMF composition. If PMF was included on the $PM_1$ samples, this would be a useful comparison.

    This is now included and moved to the main text as also suggested by Referee#2. However, it should be noted that the discussion on the PM1 composition is limited as the low flow rate used during the campaign resulted in concentrations below the detection limit, notably regarding the metal water-soluble fraction.

**Minor Comments:**

- Section 3.2.1: What factors are driving the change in the $Cl^-/Na^+$ ratio? Are the lower values observed during P1 due to acid displacement of chloride in sea salt, or do you expect non sea salt sources of these ions during different periods.

    We believe that the lower values observed in P1 are due to acid displacement of chlorine in sea salt. Acidity is elevated in this period due to the high concentrations of fluoride

- Section 3.2.2. The source of fluoride being the marine shelf is intriguing. Can the authors comment on the mechanism of how the aerosol ends up enriched in F? Does the sea water in that region have higher F content?

While we are not aware of measurements of F in the Namibian sea water, Atlas and Pytkowicz (1977) and Hossein et al (2024) describe the mechanims by which F could be released in sea water by dissolution. Upon dissolution, the release of F− to the atmosphere can be attributed to the reaction with hydrogen in water to form hydrogen fluoride gas (or a solution of hydrofluoric acid; Anbar and Neta, 1967). The high content of fluoride in the Namibian soil is also documented and attributed to weathering and dissolution of fluoride-containing minerals (Hossein et al, 2024). These comments and references are now added to the manuscript.

- Line 414: "…and with the exception of a peak value on 26 August, the Si/Al ratio…" the figure does not show a peak on this day, should this be another date?

  We thank Referee#1 for spotting out this mistake, the correct date is 24 August 2017

- Consider showing the time series of the PMF factors in the main text.

  This is now done

- Consider dividing P3 into two sections in the time series in the main text as is done in the supplemental box plots, especially Figure 4. It is clear that there are two regimes, but this is not discussed in the text until later in the manuscript.

  This is now done

- Figure 3: Please clarify in the figure captions when gaps in the graphs correspond with missing data (as is the case in figure 1) and when the measurements were below the limit of detection (as mentioned in the text for figure 3). Also, please label P1 and P2 for consistency with other figures

  This is now done

- Figure 5: Could these be labeled with the date they were collected? The four XANES spectra were chosen because they had the highest Fe loading. Do the authors think the fact that these four appear similar is due to a similar source for these four. If so, it may be more interesting to include different examples in Figure 5, such as the samples with clay/hematite signatures, or Fe(II) signatures mentioned in the text. Overlaying the spectra may also help the readers observe small differences between the spectra.

  We thank the Referee#1 for the suggestions. We added the dates and modified the figure so to address so to display a selection of samples representing different compositions, origin and periods. `We also added the display of the contributions of the standards to the deconvolution. We tried to overlay the spectra but it is difficult to show them in a clear way. So we decided not to.

- Supplemental: In some cases one of the PMF factors is called Si-rich, and others it is Sand.

  These instances are now corrected

**Typographical**

Line 416: Planes should be replaced with plains.

Line 592: Gater should be replaced with Later.

These are now corrected

Reference

Anbar, M., and Neta, P.: Reaction of fluoride ions with hydrogen atoms in aqueous solution, Transactions of the Faraday Society, 63, 141-146, 10.1039/tf9676300141, 1967.

Atlas, E., and Pytkowicz, R. M.: Solubility behavior of apatites in seawater, Limnology and Oceanography, 22, 290-300, https://doi.org/10.4319/lo.1977.22.2.0290, 1977.

Hossein, M., Rwiza, M. J., Nyanza, E. C., Bakari, R., Ripanda, A., Nkrumah, S., Selemani, J. R., and Machunda, R. L.: Fluoride contamination a silent global water crisis: A Case of Africa, Scientific African, 26, e02485, https://doi.org/10.1016/j.sciaf.2024.e02485, 2024.

---

## Author Response (AR3)

Créteil, France – 16 July 2025

Dear Editor,

Please find hereby the revised version of manuscript egusphere-2025-446 titled "Part 2: Elemental composition, iron mineralogy and solubility of anthropogenic and natural mineral dust aerosols in Namibia: a case study analysis from the AEROCLO-sA campaign".

We have modified the files to take into account your comments.

"We hope that this research paper will retain your attention for publication on Atmospheric Chemistry and Physics

With my very best regards

Paola Formenti, corresponding author